# Genome-Wide Characterization and Expression Profiling of HD-Zip Genes in ABA-Mediated Processes in *Fragaria vesca*

**DOI:** 10.3390/plants11233367

**Published:** 2022-12-04

**Authors:** Yong Wang, Junmiao Fan, Xinjie Wu, Ling Guan, Chun Li, Tingting Gu, Yi Li, Jing Ding

**Affiliations:** 1State Key Laboratory of Crop Genetics and Germplasm Enhancement, College of Horticulture, Nanjing Agricultural University, Nanjing 210023, China; 2Institute of Pomology, Jiangsu Academy of Agricultural Sciences, Jiangsu Key Laboratory for Horticultural Crop Genetic Improvement, Nanjing 210014, China; 3Department of Plant Science and Landscape Architecture, University of Connecticut, Storrs, CT 06269, USA

**Keywords:** HD-Zip, *Fragaria vesca*, ABA, fruit, expression

## Abstract

Members of homeodomain-leucine zipper (HD-Zip) transcription factors can play their roles by modulating abscisic acid (ABA) signaling in *Arabidopsis*. So far, our knowledge of the functions of HD-Zips in woodland strawberries (*Fragaria vesca*), a model plant for studying ABA-mediated fruit ripening, is limited. Here, we identified a total of 31 HD-Zip genes (*FveHDZ1-31*) in *F. vesca*, and classified them into four subfamilies (I to IV). Promoter analyses show that the ABA-responsive element, ABRE, is prevalent in the promoters of subfamily I and II *FveHDZs*. RT-qPCR results demonstrate that 10 of the 14 investigated *FveHDZs* were consistently >1.5-fold up-regulated or down-regulated in expression in response to exogenous ABA, dehydration, and ABA-induced senescence in leaves. Five of the six consistently up-regulated genes are from subfamily I and II. Thereinto, *FveHDZ4,* and *20* also exhibited significantly enhanced expression along with increased ABA content during fruit ripening. In yeast one-hybrid assays, FveHDZ4 proteins could bind the promoter of an ABA signaling gene *FvePP2C6*. Collectively, our results strongly support that the *FveHDZs*, particularly those from subfamilies I and II, are involved in the ABA-mediated processes in *F. vesca*, providing a basis for further functional characterization of the HD-Zips in strawberry and other plants.

## 1. Introduction

In plants, transcription factors (TFs) have been shown to regulate nearly all biological processes. They can bind to conserved and specific promoter regions (e.g., *cis*-acting elements) to promote or repress the mRNA transcriptional levels of their target genes [1,2]. *Homeobox* (*HB*) genes are a superfamily of TFs existing in all eukaryotic organisms, and all of the HB proteins contain a homeodomain (HD) that is a conserved 60-amino acid motif [3]. Members of the homeodomain-leucine zipper (HD-Zip) family are characterized by a leucine zipper (LZ) motif immediately adjacent to the HD, which is unique to plants [4]. In HD-Zip proteins, the HD is demonstrated to be responsible for the specific binding to DNA, while the LZ acts as a dimerization motif [4,5].

So far, HD-Zips have been identified and analyzed in several plants, e.g., *Arabidopsis* [4], poplar [6], and rice [7]. Based on their evolutionary relationships and structural characteristics, the HD-Zip family is divided into four subfamilies (HD-Zip I, II, III, and IV) [4]. Studies show that each of the four HD-Zip subfamilies has its own specific protein structures [4,8]. In general, HD-Zip I members only have HD and LZ domains, while there is another motif, named CPSCE in HD-Zip II members. A START domain with putative lipid binding capability is especially found in both HD-Zip III and IV proteins. The most distinguishable feature for HD-Zip III and IV proteins is that HD-Zip III members own a distinctive MEKHLA domain at protein C-terminal, while two LZ motifs adjacent to the HD, which form a leucine zipper with an internal loop (a.k.a. ZLZ motif), exist in HD-Zip IV protein sequences [8].

HD-Zips are demonstrated to be involved in many biological processes, and the members from different subfamilies have distinct roles in plants [4,8]. For instance, HD-Zip I genes have been implicated in the de-etiolation of dark-grown seedlings [9] and abiotic stress responses and tolerance, particularly drought, salinity, and cold stress [10,11]. Some studies also indicate that HD-Zip I genes can be involved in the regulation of plant senescence [12,13], flowering [14] and fruit ripening [15,16]. Besides, *LeHB-1* of HD-Zip I in tomato is thought to positively mediate the regulation of the climacteric fruit ripening [15]. HD-Zip II genes are thought to be mainly in response to illumination conditions [17], auxins [18], and shade-avoidance [19]. One HD-Zip II gene has been reported to positively regulate peach fruit ripening [16]. HD-Zip III genes are mainly involved in embryogenesis [20], meristem function [21,22], and vascular development [20,23]. HD-Zip IV genes are shown to be mainly involved in epidermal cell differentiation, anthocyanin accumulation, root development, and trichome formation as well as reproductive growth [24,25,26].

Abscisic acid (ABA) is a very important plant hormone, and has been shown to regulate many aspects of plant growth and development including leaf senescence, seed dormancy, embryo maturation, and responses to drought and other environmental stresses [27,28,29]. Several genes from HD-Zip I are demonstrated to be involved in the ABA signaling pathway and in the regulation of plant developmental adaptation to environmental stresses [11]. These genes include *ATHB6*, *7,* and *12* from *Arabidopsis* [12,30,31,32,33], and *Hahb4* from sunflower [34,35]. ATHB6 protein has been demonstrated to negatively regulate the ABA signaling pathway by interacting with the protein phosphatase 2C gene (*PP2C*) *ABI1* [31]. Additionally, ABA-induction of *ATHB6*, *7,* and *12* was abolished in the ABA insensitive mutants *abi1* and *abi2* [30,31,33], and the *athb7 athb12* double mutant lost the capacity to respond to increases in ABA concentration [32], all of which indicated that these HD-Zip genes function in an ABA-dependent signaling pathway.

Besides an important abiotic stress hormone, ABA is also essential for strawberry fruit ripening [36]. Thus, identification and analysis of ABA signaling genes are very meaningful in strawberry. Up to now, no detailed information is available regarding the features of the HD-Zip gene family and the responses of HD-Zips to ABA in strawberry. Woodland strawberry (*Fragaria vesca*) is considered a model plant for the study of octaploid strawberry (*Fragaria* × *ananassa*) and other non-climacteric fruits, because of the small genome (about 240 M), short growth cycle, and many other advantages [37,38,39,40]. In this study, we first performed a genome-wide identification and phylogenetic analysis of HD-Zip family genes in *F. vesca*. We then characterized their structures and expression patterns during fruit development and ripening, and in response to exogenous ABA, ABA-induced senescence and dehydration in leaves. The possible interaction of FveHDZ4 and FveHDZ20 proteins with the promoters of *FvePP2C* genes was further tested by yeast one-hybrid assays. Collectively, our results provide an insight into HD-Zip family genes in woodland strawberry and lay a foundation for further studies on their roles in ABA-mediated processes.

## 2. Results

### 2.1. Identification of HD-Zip Family Members in F. vesca

To identify the HD-Zip family members in *F. vesca*, we performed HMMER searches in the *F. vesca* proteome using the full alignment of the HD domain downloaded from Pfam as query sequences (Materials and Methods). The matching proteins were further submitted to the MEME website for identifying the LZ motifs. As a result, 31 proteins, which contained both HD domains and LZ motifs, were regarded as the HD-Zip family members in *F. vesca* (Table 1 and Appendix A). We used these proteins (named FveHDZ1-31) for subsequent analyses in this study. Their predicted grand averages of hydropathicity (GRAVY) are all below zero (Table 1), suggesting that they are hydrophilic proteins. Subcellular localization analysis indicates that all the FveHDZ proteins are located in the nucleus (Table 1), which is in agreement with the fact that they are TFs.

### 2.2. Phylogenetic and Motif Analyses of FveHDZ Proteins

To study the evolutionary relationships among these FveHDZs, an unrooted neighbor-joining phylogenetic tree was generated based on protein sequences of the 31 FveHDZs using the MEGA10 software. As shown in Figure 1, the 31 FveHDZs are well clustered into four subfamilies, namely subfamily I-IV. To test the validity of this classification, another phylogenetic tree based on the HD-Zip protein sequences in *F. vesca*, *Arabidopsis*, grape, and peach was constructed using the same method (Appendix A). The clustering of the FveHDZs and the known *Arabidopsis* HD-Zips from the four subfamilies [4] confirms the grouping of the 31 FveHDZs in Figure 1. There are 13 (FveHDZ1-13), 7 (FveHDZ14-20), 4 (FveHDZ28-31) and 7 (FveHDZ21-27) proteins in subfamily I-IV, respectively (Figure 1). Protein sequence analysis shows that closely related FveHDZs in the same subfamilies have similar lengths (Table 1). In addition, the FveHDZs from subfamily III (838–849 aa) and IV (709–829 aa) are significantly longer than the ones from subfamily I (172–329 aa) and II (209–340 aa).

Using the MEME suite, a total of 18 conserved motifs were identified in the 31 FveHDZs (Figure 1 and Appendix A). As expected, motifs 1 and 2 that correspond to the HD domain, as well as motif 3 equal to the LZ motif, were found in all the FveHDZs. The other motifs only existed in one or two subfamilies. For example, motifs 4, 5 and 6 corresponding to the START domain were merely present in subfamily III and IV; motif 10 specifying the MEKHLA domain was specific to subfamily III, while motif 18 corresponding to the CPSCE motif was found only in subfamily II and III. Besides, similar to the *Arabidopsis* HD-Zips in subfamily IV [4], the FveHDZs in subfamily IV characteristically contained two LZ motifs adjacent to the HD domain. These results demonstrate that the FveHDZs in the same subfamily share a common motif composition, which supports the classification of the 31 FveHDZs from our phylogenetic tree.

### 2.3. Chromosomal Location, Gene Structure, and Synteny Analyses of FveHDZs

Exon-intron structures and chromosomal locations of the 31 *FveHDZs* were retrieved from the genome annotation file of *F. vesca* (Table 1 and Figure 1). The results show that the members within the same subfamily generally share similar gene structures and intron numbers. Besides, the introns of *FveHDZs* from subfamily III (17) and IV (9 to 11) are much more than those from subfamily I (0 to 3) and II (2 to 3). Most chromosomes contain 3–5 *FveHDZs* from different subfamilies, while chromosome LG6 harbors the most (7), and chromosome LG2 harbors the least number (2) of the genes (Appendix A and Figure 2).

Additionally, syntenic regions in the *F. vesca* genome were analyzed by the MCScanX software [41], and possible evolutionary relationships among the 31 *FveHDZs* were identified. The results showed that thirteen of them were derived from WGD/segmental duplication, one was from proximal duplication, and the others were dispersed genes (Table 1). The thirteen WGD/segmental duplicate *FveHDZs* comprised seven duplicate pairs of which four, two, and one pairs belonged to subfamily I, II, and IV, respectively (Table 1 and Figure 2). No WGD/segmental duplication event was found in subfamily III. These results suggest that WGD/segmental duplication was the main mechanism for the expansion of *FveHDZs* which we could trace.

### 2.4. Promoter Analysis of FveHDZs

To investigate the possible regulatory mechanism of *FveHDZs*, 1.5 kb promoter sequences upstream of the transcription start sites (ATGs) were extracted and used for *cis*-element prediction. As a result, 48 different *cis*-elements were identified and could be divided into four categories: 11 phytohormone response related, 6 abiotic and biotic stress response related, 7 plant growth and development related, and 24 light response-related *cis*-elements (Appendix A). Among these *cis*-elements, ABRE which is regarded as an ABA response element [42] occurred the third most frequently (46 times, Appendix A) in the *FveHDZ* promoters. Particularly, more than 80% of the ABREs were found in subfamily I (29/46) and II (9/46). A total of eight *FveHDZs*, including four from subfamily I, three from II, and 1 from IV, had more than three ABREs predicted in their promoters, where *FveHDZ4* from subfamily I contained the most ABREs (8). These results suggest that the *FveHDZs*, especially those from subfamily I and II, are likely involved in the ABA-mediated processes of *F. vesca*.

### 2.5. Expression Analysis of FveHDZs during Fruit Development and Ripening

We then investigated expression profiles of the *FveHDZs* in different ABA-mediated processes, to figure out whether and which members of the family play a role in these processes. First, the fruit ripening process. We downloaded the transcriptome data of *F. vesca* fruits from the open online resources (Materials and Methods), and analyzed *FveHDZ* expression levels in various developmental and ripening tissues/stages. During achene development, all 31 *FveHDZs* except *FveHDZ12* were expressed (Figure 3a). Most *FveHDZs* in subfamily I and II, such as *FveHDZ2* and *16*, were mainly expressed in tissues other than the embryo and had larger expression changes among the tissues than among the stages. The expression levels of *FveHDZ3*, *4*, *6*, and *16* gradually increased, while those of *FveHDZ7*, *14*, and *18* decreased, during both ghost and wall development. Different from the other members, *FveHDZ5* was expressed specifically in the embryos with its peak at stage 4 (the torpedo-shaped embryo, Figure 3a), suggesting that *FveHDZ5* may play a role in the embryo development of *F. vesca*. In contrast, all the *FveHDZs* in subfamily III and IV except *FveHDZ23* were highly expressed in every achene tissue, with small expression changes among the tissues/stages (Figure 3a). Together, these results demonstrate that the *FveHDZs* from subfamily I and II have diversified expression patterns, while those from subfamily III and IV have a constitutive expression pattern, in achene development.

During receptacle development from stage 1 to stage 5, all the *FveHDZs* in subfamily IV were preferentially expressed in the cortex, while a majority of *FveHDZs* (17/24) from subfamily I-III showed similar expression levels in both cortex and pith (Figure 3b). In the ripening stages of the receptacle from SW to Red, most *FveHDZs* (19/31) exhibited decreased expression levels (Figure 3c). Among the remaining 12 *FveHDZs*, 5 were highly expressed throughout the ripening stages, while 3 (*FveHDZ4*, *17*, and *20*) from subfamily I and II displayed an increasing trend. Particularly, the expression of *FveHDZ4* and *20* was greatly enhanced at the Pre-T stage when the ABA content starts to dramatically boost in receptacles and the fruits transit from immature to ripe [45]. This suggests that these two *FveHDZs* may be induced by ABA in the receptacles and play roles in the ripening of *F. vesca* fruits.

### 2.6. Expression Analysis of FveHDZs in Response to ABA

Second, the rapid response to ABA. Based on the above promoter and expression analyses, we selected 18 *FveHDZs*, including 8, 5, 1, and 4 from subfamily I, II, III, and IV, respectively, for this and the following expression analyses. In this experiment, exogenous ABA was applied to detached leaves, and expression of the 18 *FveHDZs* was examined at 0, 3, 6, and 12 h. As shown in Figure 4, 11 and 5 genes had more than 1.5-fold up-regulated and down-regulated expression, respectively, in response to the ABA treatment. Among them, 12 genes already exhibited significant expression changes at 3 h compared with 0 h, e.g., *FveHDZ4* and *11*, whose expression varied remarkably and reached their top and bottom levels, respectively (Figure 4). These results indicate that these *FveHDZs* are able to respond to ABA very rapidly. On the other hand, 9 of the 11 ABA-induced *FveHDZs* were from subfamilies I and II. In other words, approximately 70% (9/13) of the studied *FveHDZs* in subfamily I and II had significantly increased expression within the 12 h ABA treatment. This suggests that most *FveHDZs* in subfamily I and II can be rapidly induced by ABA and participate in the downstream signaling pathways.

### 2.7. Expression Analysis of FveHDZs in ABA-Induced Leaf Senescence

Third, the ABA-induced leaf senescence. We incubated detached *F. vesca* leaves in dark with 100 μM ABA or 0.02% DMSO solvent (without ABA, as a control), and found that the addition of ABA greatly accelerated the yellowing of the leaves (Appendix A). Moreover, the marked increase in expression of the senescence marker gene *FveSAG12* was observed at 1 d and 5 d after the incubation with and without ABA, respectively (Appendix A). This indicated that leaf senescence occurred approximately four days earlier with ABA than in the control, and ABA significantly promoted the senescence process of these detached leaves in dark.

Accordingly, we measured the expression levels of the 18 selected *FveHDZs* in the incubated leaves from 0 d to 5 d. The RT-qPCR results showed that 15 *FveHDZs* had significant expression differences between the leaves incubated with and without ABA (Figure 5). Thereinto, eight genes, seven of which were from subfamily I and II, exhibited significantly increased expression in the ABA-treated leaves than in the control. Further, these expression increases could be clearly detected after a 1-day incubation with ABA in six of the eight *FveHDZs*, e.g., *FveHDZ1*, *4*, and *20*, similar to in *FveSAG12*. This strongly suggests that these six genes played a role in ABA-induced leaf senescence. In contrast, seven genes from all the four subfamilies, such as *FveHDZ11*, *15*, *22*, and *30*, demonstrated significantly reduced expression during the 5-day incubation with ABA. Their expression was either increased or unaltered in the control leaves, suggesting that these genes were repressed in the ABA-induced leaf senescence.

### 2.8. Expression Analysis of FveHDZs in Response to Dehydration

Forth, leaf dehydration. It has been widely reported that drought stress leads to rapid increase of endogenous ABA content in leaves [46]. In this experiment, we placed the detached *F. vesca* leaves in boxes for dehydration (Materials and Methods). To test whether this treatment was effective, we examined water loss from the leaves and expression of *FveNCED1*, the key ABA synthesis gene in response to drought stress [47], from 0 h to 12 h. The results showed that the water loss already reached about 20% at 1 h (Figure 6a). Meanwhile, the *FveNCED1* expression had been strongly induced (Figure 6b), strongly supporting that the leaves were dehydrated, and endogenous ABA was rapidly synthesized.

We next investigated the expression of 14 of the above 18 *FveHDZs* which exhibited differential expression patterns among each other in the above analyses. As shown in Figure 6c, 13 of these 14 *FveHDZs*, except *FveHDZ1*, had more than 1.5-fold expression changes during the 12-h dehydration. Among them, only expression of *FveHDZ3*, *4*, and *20* was markedly enhanced within 6 h and remained significantly higher than 0 h during the following period (Figure 6c). Expression of the other three dehydration-induced *FveHDZs* all transiently increased and fell back to the control levels (0 h) at the end of the experiment. On the other hand, there were seven *FveHDZs* demonstrating significantly decreased expression during dehydration. Particularly, expression levels of *FveHDZ11*, *22,* and *26* were strongly reduced to less than 20% of control at 6 h, suggesting that they were repressed in response to the increase in water loss and/or ABA content in the leaves. Overall, these results indicated that most studied *FveHDZs* were likely involved in the dehydration-induced responses in the detached leaves.

### 2.9. Binding of FveHDZs to the Promoter Fragments of Protein Phosphatase 2C Genes (PP2Cs) in Y1H Assays

It has been demonstrated that two ABA-induced HD-Zip proteins (ATHB7 and 12) are negative regulators of ABA signaling by activating *PP2C* genes in *Arabidopsis* [32]. To test whether FveHDZs can interact with the promoters of *FvePP2C* genes in *F. vesca*, we selected two *FveHDZs*, *FveHDZ4* and *20*, which exhibited consistently and highly increased expression in all the above analyses (Figure 7) for the Y1H assays. Two *PP2C* genes, *FvePP2C6* and *FvePP2C1*, whose expression patterns were similar to that of *FveHDZ4* during fruit ripening (Appendix A) were selected as the candidate target genes. Two fragments (FvePP2C6-F1 and FvePP2C6-F2) from the *FvePP2C6* promoter and one fragment (FvePP2C1-F1) from the *FvePP2C1* promoter, which contained the known HD-Zip-binding sites, were used in the Y1H assay (Figure 8a,b, and Appendix A). As a result, the assay between FveHDZ4 and the promoter fragment FvePP2C6-F2 showed possible interaction (Figure 8c), suggesting that FveHDZ4 may have the ability to bind the promoter of *FvePP2C6*.

## 3. Discussion

In this study, a comprehensive analysis of HD-Zip family genes was performed in *F. vesca*. A total of 31 *FveHDZs* were identified and classified into four subfamilies, I-IV, corresponding to the subfamilies HD-Zip I-IV in *Arabidopsis*, respectively (Appendix A). Our following analyses revealed that gene structure, as well as motif/domain composition and protein length of the FveHDZs, exhibited subfamily-specific patterns. Moreover, these patterns are mostly consistent with the previous reports in other plants, such as *Arabidopsis* [4], rice [7], and peach [48], providing further evidence for our classification of the *FveHDZs*.

The number of *FveHDZ* genes is much smaller than the HD-Zip numbers in *Arabidopsis* (48)[6], poplar (63)[6], rice (48)[7], and maize (55)[49] (Appendix A). MCscanX analysis suggested that the expansion of *FveHDZs* may be mainly resulted from WGD/segmental duplication. Compared with the above three plant species, *F. vesca* did not undergo any recent WGD duplication event after the divergence of core eudicots (Appendix A) [50], which may account for the smallest number of HD-Zip genes in its genome. Grape and peach, which contain similar numbers of HD-Zips with *F. vesca* (Appendix A), did not undergo recent WGDs, either [50]. Additionally, Of the 13 WGD/segmental duplicate *FveHDZs*, 6 have orthologous grape *HD-zips* (Appendix A, Appendix A) located in the syntenic regions in the grape genome [51]. The latest WGD event shared by *F. vesca* and grape was proposed to have occurred in the common ancestor of all core eudicots [50]. Therefore, those orthologous WGD/segmental duplicate HD-Zips may likely be traced back to this ancestral WGD event. Taken together, the consistency between the HD-Zip number and the occurrence of recent WGD(s) supports the idea that WGD/segmental duplication was the main mechanism for the HD-Zip expansion in eudicots.

ABA is a central hormone in controlling fruit ripening and other developmental and stress processes in strawberries [36]. The genes from HD-Zip I in *Arabidopsis* have been shown to play an important role in the ABA signaling pathway [4,8,32]. However, whether the *FveHDZs* from subfamily I and other subfamilies are involved in ABA signaling and responses remains unclear. Our RT-qPCR results demonstrate that 10 of the 14 investigated *FveHDZs* (71.4%) were consistently >1.5-fold up-regulated or down-regulated in expression in the three ABA-mediated processes in leaves, including rapid ABA response, dehydration, and ABA-induced senescence (Figure 7). Analyses of the online transcriptome data of *F. vesca* receptacles reveal that five of these ten *FveHDZs* also exhibited similar expression changes along with the increased ABA content during fruit ripening. Therefore, it is likely that these five *FveHDZs*, including *FveHDZ4* and *11* from subfamily I and *FveHDZ20*, *22,* and *30* from subfamily II, IV, and III, respectively, are involved in ABA responses in *F. vesca*.

Of these five *FveHDZs*, two genes (*FveHDZ4* and *20*) were consistently up-regulated in all four investigated processes, whereas the remaining three genes were down-regulated. This indicates that these two kinds of *FveHDZs* may be modulated by different upstream TFs and meanwhile modulate different downstream genes in the ABA signaling pathway. The other investigated *FveHDZs* exhibited diverse expression changes among leaves and fruits, or even among the three processes in leaves, suggesting that their expression is regulated in a tissue- and/or process-specific manner. It would be interesting to find out whether their roles in the control of downstream gene expression are different among the tissues/processes or not in future studies. Overall, our expression profiling of the *FveHDZ* genes provides a clue for characterizing the comprehensive scene of the genetic mechanism of ABA responses in terms of the HD-Zip gene family.

Previous studies in climacteric fruits have revealed that the *LeHB-1* (HD-Zip I gene) in tomato and the *PpHB.G7* (HD-Zip II gene) in peach positively regulates fruit ripening [15,16]. *F. vesca* is a model plant for studying non-climacteric fruit ripening [37]. According to the transcriptome analyses, *FveHDZ4* and *20* were strongly up-regulated at the Pre-T stage of the ripening fruits and then highly expressed thereafter (Figure 3c). The Pre-T stage was considered to be the transition stage from immature to ripe fruits in *F. vesca*, and the ABA content in *F. vesca* fruits exhibited a very high increase since the Pre-T stage [45]. This concordance between *FveHDZ4/20* expression and ABA content suggests that *FveHDZ4* and *20* may be induced by the increased ABA levels, and play a role during the fruit ripening process. Altogether, our results provide evidence that the HD-Zip genes also have regulatory functions in the ripening of non-climacteric fruits.

Leaf dehydration and senescence are two typical processes mediated by ABA [27,29]. As important TFs induced by ABA, many HD-Zip proteins have been shown to participate in the dehydration/drought and senescence processes as well. For instance, the ABA-induced HD-Zip genes in *Arabidopsis*, *ATHB6*, *7* and *12*, could also be rapidly up-regulated by drought [30,52,53]; and *ATHB6* overexpression can enhance the drought tolerance of maize [53]. Moreover, *ATHB7* [12], as well as the ABA-induced HD-Zips in other plants, e.g., *Oshox4* [7], *Hahb-4* [54,55], and *CsHB5* [13], are involved in plant senescence. Our experiments confirm the important roles of ABA in leaf dehydration and senescence in *F. vesca*. Expression of the key ABA synthesis gene *FveNCED1* was strongly and rapidly enhanced during leaf dehydration, and continuous ABA treatment markedly accelerated leaf senescence. Correspondingly, 6 of the 14 investigated *FveHDZs*, such as *FveHDZ4* and *20*, were significantly up-regulated in expression. From the phylogenetic analyses, *FveHDZ4* and *20* are the orthologs of *ATHB7* and *12* (Appendix A). The consistency in their expression patterns in response to ABA and dehydration suggests that *FveHDZ4* and *20* likely have similar roles with *ATHB7* and *12* in these processes.

ABRE has been regarded as a major *cis*-acting regulatory element in ABA-dependent gene expression [42]. Our promoter analysis indicates that ABRE is the third most abundant *cis*-element detected in the promoters of the 31 *FveHDZs*. RT-qPCR results demonstrate that 12 of the 14 ABRE-containing *FveHDZs* could respond to the ABA treatment in a short time (Figure 7). Additionally, the prevalence of ABREs in the promoters of subfamily I and II *FveHD*Zs is in accordance with the significantly enhanced expression of most of these genes upon the ABA treatment of leaves (Figure 7). *FveHDZ4*, whose promoter contains the most ABREs, displayed the most strongly and rapidly responses. Although counter-examples existed, our results support the idea that multiple ABREs contribute to a rapid ABA response [56].

The clade A *PP2C* genes play a central role in the ABA-perception mechanism, acting as negative regulators of ABA signaling [57]. In *Arabidopsis*, two HD-Zip proteins (ATHB7 and ATHB12) have been demonstrated to bind with *PP2C* promoters, and positively regulate the genes [32]. The *athb7athb12* double mutant lost the capacity to respond to increases in ABA content, which is similar to the *pp2c* triple mutants, indicating that HD-Zip genes are important for the maintenance of ABA sensitivity in plants [32]. The Y1H assay in this study suggests that FveHDZ4 may have the ability to directly bind the *FvePP2C6* promoter. Expression of *FveHDZ4* and *FvePP2C6* exhibits consistently similar variation patterns among the tissues/stages during fruit development and ripening (Appendix A, correlation coefficient = 0.896 between expression levels of *FveHDZ4* and *FvePP2C6* shown in Appendix A), supporting that the *FvePP2C6* expression may be induced by FveHDZ4. These results provide preliminary evidence for the regulatory role of FveHDZ4 in modulating the *FvePP2C6* expression. Further experiments are needed to confirm the interaction of FveHDZ4 and the *FvePP2C6* promoter, and to figure out whether the synchronously enhanced expression of *FveHDZ4* and *FvePP2C6* is common in all ABA-mediated processes. These future studies may help us to understand the role of FveHDZ4 and other FveHDZs in the regulation of *FvePP2C* genes and ABA signaling in *F. vesca*.

Another interesting observation is that *FveHDZ5* was almost uniquely expressed in the embryos from stage 3 to stage 5 according to the transcriptome data analysis in *F. vesca* fruits. According to the previous description [40], the seed at stage 3 contains a heart stage embryo, while at stage 5, the two cotyledons of the embryo stay upright and fill the entire seed, indicating the maturation of the embryo. Our promoter analysis shows that there is an RY-element present in the promoter of *FveHDZ5* (Appendix A). Studies have demonstrated that the RY-element mainly exists in the promoters of seed-specific genes and is important for their expression during plant embryo development [58,59]. Consequently, it is likely that *FveHDZ5* is an embryo-specific gene and may play a role in the embryo development of woodland strawberry.

## 4. Materials and Methods

### 4.1. Identification and Analysis of HD-Zip Family Proteins in F. vesca

The whole genome and protein sequences of *F. vesca* (v4.0.a2) were obtained from the Genome Database for Rosaceae (GDR, https://www.rosaceae.org, accessed on 23 December 2021). If there were more than one protein annotated for the same gene because of alternative splicing, the longest form was used for our analysis. Full-alignment sequences of the homeobox domain (HD, PF00046) were used as queries to identify the putative HD proteins in *F. vesca* by the local HMM-based searches using HMMER3 [60]. Domains presented in the target proteins were further validated by Pfam35.0 (http://pfam.xfam.org/, accessed on 28 December 2021), the Simple Modular Architecture Research Tool (SMART; http://smart.embl-heidelberg.de/, accessed on 28 December 2021), and the Conserved Domain Database (CDD; http://www.ncbi.nlm.nih.gov/Structure/cdd/wrpsb.cgi, accessed on 28 December 2021) available from NCBI, with a threshold of E-value < 1 × 10^−5^. According to previous reports [61,62], HD proteins in *F. vesca* containing the characteristic domains of other family genes (e.g., PHD, POX, KNOX, DDT) were removed. The remaining HD proteins were then submitted to the MEME suite 5.4.1 (http://meme-suite.org/tools/meme, accessed on 6 January 2022) to identify the LZ (leucine zipper) motif. The proteins containing both the HD domain and LZ motif were regarded as HD-Zip family proteins in *F. vesca*.

The conserved motifs in HD-Zips were further predicted using the MEME program with the following parameters: number of repetitions, any; the maximum number of motifs, 18 and the optimum motif widths, between 5 and 50 residues, and the E-value of each motif was set to be less than 1 × 10^−5^. The predicted motifs were further analyzed by searching against the InterProScan database (http://www.ebi.ac.uk/Tools/pfa/iprscan/, accessed on 12 January 2022). Biochemical parameters of HD-Zip proteins were calculated by the ExPASy database (https://web.expasy.org/protparam/, accessed on 13 January 2022), and subcellular parameters of them were predicted by the Plant-mPLoc (http://www.csbio.sjtu.edu.cn/bioinf/plant-multi/, accessed on 13 January 2022).

### 4.2. Phylogenetic Analysis and Gene Nomenclature

The 48, 33, and 33 HD-Zips from *Arabidopsis*, peach, and grape, respectively, were obtained from previous papers [6,48,51]. Multiple sequence alignments of the full-length protein sequences from different species were performed by MUSCLE v3.8.31 with default parameters [63]. Then, MEGA10 was used to construct the phylogenetic tree using the neighbor-joining method with the Jones-Taylor-Thornton (JTT) model [64]. A total of 1000 bootstrapping replicates were analyzed with the partial deletion model for gaps/missing data.

The HD-Zips from *F. vesca* were named by arabic numerals following the *F. vesca* abbreviation. For convenience, a three-letter abbreviation (Fve) was used according to the nomenclature system proposed by the Rosaceae research community [65]. The accession number of each HD-Zip gene in *F. vesca* (referred to as *FveHDZ*) was listed in Appendix A.

### 4.3. Gene Structure, Chromosomal Location, and Gene Duplication

The exon/intron organization and chromosomal location for the individual *FveHDZ* were achieved from the genomic and transcriptional annotation information obtained from the GDR. The schematic diagram of chromosomes was drawn by MapInspect software. The possible duplication mechanisms were analyzed by MCscanX [41].

### 4.4. Identification of Putative Promoter Cis-Acting Elements

*Cis*-acting elements located in the *FveHDZ* promoters were analyzed from the 1500 bp regions upstream of the transcription start sites using the PlantCARE database (http://bioinformatics.psb.ugent.be/webtools/plantcare/html/, accessed on 22 January 2022).

### 4.5. Plant Materials and Stress Treatments

Seeds of *F. vesca* (inbred line Hawaii 4, a gift from Dr. Janet Slovin) were surface-sterilized and grown in MS media as described by Wang et al. [47]. Well-expanded leaves cut from 50-day-old seedlings were used in the following treatments. All the samples were immediately put into liquid nitrogen for RNA extraction after treatments.

In the experiment of rapid responses to ABA, the leaves were first put into sterile water containing 0.02% (*v*/*v*) dimethyl sulfoxide (DMSO, used for the dissolution of ABA), and then shaken gently (100 rpm/min) overnight at 22 °C in dim light. On the second day, leaves were transferred to a new bottle with 100 μM ABA in 100 mL water for ABA treatment, and collected at 0 h, 3 h, 6 h, and 12 h, respectively.

In the ABA-induced senescence experiment, the leaves were placed in sterile plates with 30 mL water. A total of 100 μM ABA or 0.02% (*v*/*v*) DMSO solvent (as a vehicle control) was added to the leaves. All plates were put into a dark environment at 22 °C. The leaves with ABA were collected at 0 d, 1 d, 3 d, and 5 d, while the leaves without ABA were collected at 0 d, 1 d, 3 d, 5 d, 7 d, 9 d, 11 d, respectively.

For the dehydration treatment, the leaves were first placed in boxes without covers at 22 °C for 60% humidity under dim light. After 1 h, the water loss of leaves reached about 20%, and the boxes were covered with plastic films to slow down further dehydration. Dehydrated samples were collected at 1 h, 6 h, 12 h, respectively, and leaves without treatment (0 h) were used as control.

### 4.6. Transcriptome Retrieve

Transcriptome data of *FveHDZs* in different fruit tissues/developmental stages was downloaded from http://bioinformatics.towson.edu/strawberry/ (accessed on 3 February 2022) [43,44,66]. Transcriptome data of *FveHDZs* in receptacles at different ripening stages were obtained from Gu et al. [45]. It should be noted that, the receptacles of *F. vesca* fruits contain two tissues: pith and cortex. The achenes on receptacles, which were usually called strawberry seeds, can be further divided into three parts: ovary wall, ghost, and embryo [40].

### 4.7. Yeast One-Hybrid Assay

The primer pairs FvePP2C6-F1_F and FvePP2C6-F1_R, FvePP2C6-F2_F and FvePP2C6-F2_R, FvePP2C1-F1_F and FvePP2C1-F1_R were used to amplify the promoter fragments FvePP2C6-F1, FvePP2C6-F2, and FvePP2C1-F1 (Appendix A), respectively, and the products were inserted into the *pAbAi* vector (Clontech). The primer pairs FveHDZ4-CDS_F and FveHDZ4-CDS_R, FveHDZ20-CDS_F, and FveHDZ20-CDS_R were used to amplify the coding sequences of *FveHDZ4* and *FveHDZ20*, respectively, and the products were ligated into the *pGADT7* vector (Clontech). The yeast one-hybrid (Y1H) assay was conducted using the MatchmakerTM Gold Yeast One-Hybrid Library Screening System (Cat.no. 630491, Clontech). All primers used were listed in Appendix A.

### 4.8. RNA Extraction, RT-qPCR Analysis, and the Selection of FveHDZs

Total RNA of each sample was extracted with a modified CTAB method [67]. Primerscript RT reagent Kit with gDNA Erase (Takara) was used to generate cDNA according to the manufacturer’s protocol. RT-qPCR reactions were carried out with SYBR Premix Ex Taq II (Takara) on a Bio-Rad CFX96. The relative expression levels were analyzed with the ΔΔCT method using *FveCHC1* as the reference gene [68]. Three biological and three technical replicates were performed for RT-qPCR. All of the specific primers we used for RT-qPCR were listed in Appendix A. The specificity and amplification efficiency of the RT-qPCR primers were validated through PCR amplification and the melting curves generated by the RT-qPCR system during the reactions.

## 5. Conclusions

In conclusion, we performed a comprehensive analysis of HD-Zip family genes in *F. vesca*, including phylogeny, gene and protein structure, expansion mechanism, expression during fruit development and ripening, and responses to different ABA-related treatments. Our results demonstrate that there are 31 *FveHDZs* in the genome and WGD/segmental duplication mainly contributes to the expansion of *FveHDZs*. The 31 *FveHDZs* are well classified into four subfamilies, and each subfamily has unique structural characteristics. ABREs are the third most abundant *cis*-elements for *FveHDZs*, and mainly distributed in the promoters of subfamily I and II genes. RT-qPCR results show that the *FveHDZs*, especially those from subfamilies I and II, could be significantly induced by ABA. Noticeably, *FveHDZ4* and *FveHDZ20* were consistently strongly and rapidly up-regulated in all the four ABA-mediated processes investigated in this study, including ABA response, ABA-induced leaf senescence, dehydration, and fruit ripening. Y1H results suggest that FveHDZ4 could bind the promoter of an ABA signaling gene *FvePP2C6*. Taken together, our findings indicate an important regulatory role of *FveHDZs* in ABA signaling and responses in *F. vesca* and would facilitate the functional characterization of HD-Zip family genes in strawberry and other plants.

## Figures and Tables

**Figure 1 plants-11-03367-f001:**
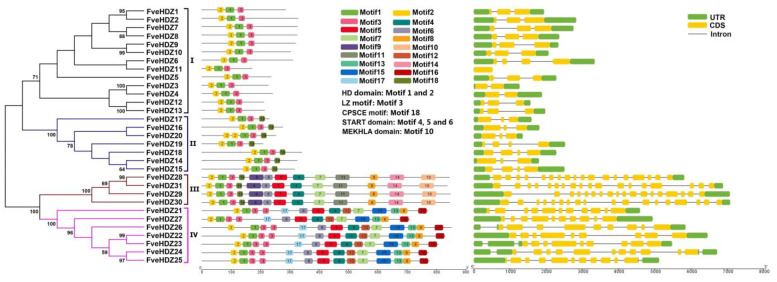
Evolutionary relationships (**left**), conserved motifs (**middle**), and gene structures (**right**) of HD-ZIP family genes in *F. vesca*. The unrooted neighbor-joining phylogenetic tree was constructed using MEGA10 based on the full-length sequences of the 31 FveHDZ proteins. Bootstrap values > 50 are shown. Genes in the subfamilies I-IV are indicated by different color blocks and lines in the phylogenetic tree. The motifs, numbered 1–20, were displayed in different colored boxes.

**Figure 2 plants-11-03367-f002:**
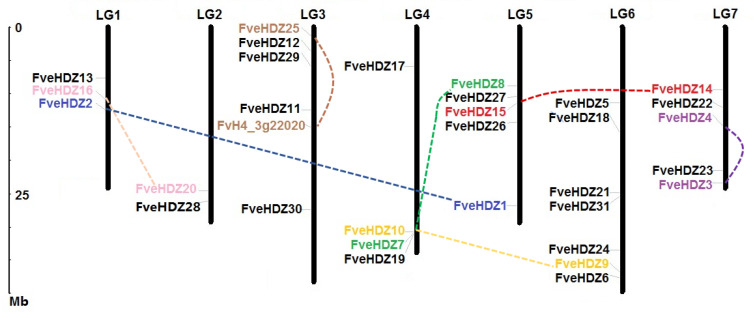
Physical locations of *FveHDZs* on *F. vesca* chromosomes. Chromosome numbers were indicated at the top of each chromosome. The scale at the left represented the length of the chromosome. Seven pairs of *FveHDZs* connected by dash lines with different colors resulted from WGD/segmental duplication. *FvH4_3g22020* was classified to be a WGD/segmental duplicate gene with *FveHDZ25* by the MCscanX analysis, but was not included in the HD-Zip family as its coding protein did not contain the HD domain and LZ motifs in the N terminus (Appendix A).

**Figure 3 plants-11-03367-f003:**
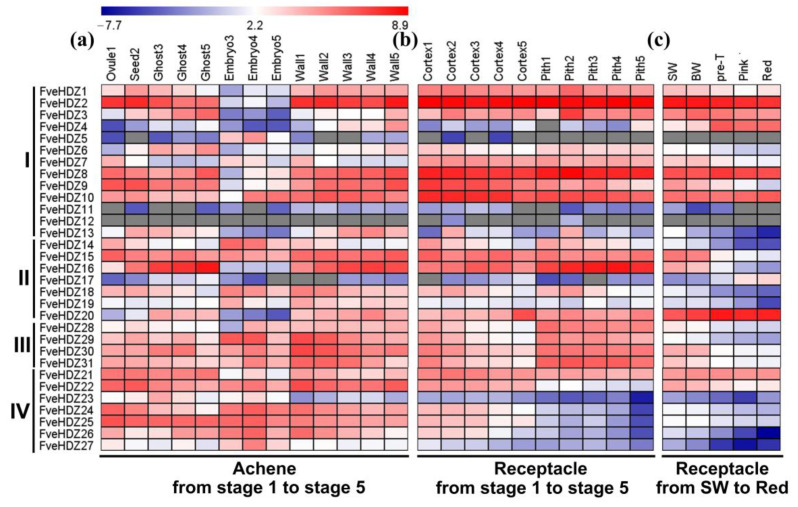
Expression patterns of *FveHDZs* in achene development (**a**) and receptacle development (**b**) and ripening (**c**). (**a**,**b**) Achenes included the ovule, seed, embryo, ghost, and wall tissues, and receptacles included the cortex and pith tissues. Numbers after the tissues indicated different fruit development stages. Data from Kang et al. [43] and Hollender et al. [44]. (**c**) SW, small white; BW, big white; and Pre-T, pre-turning. Data from Gu et al. [45]. The heat map showed log2 “relative RPKM values” of individual *FveHDZ* genes. Gray boxes indicated undetectable expression of the genes.

**Figure 4 plants-11-03367-f004:**
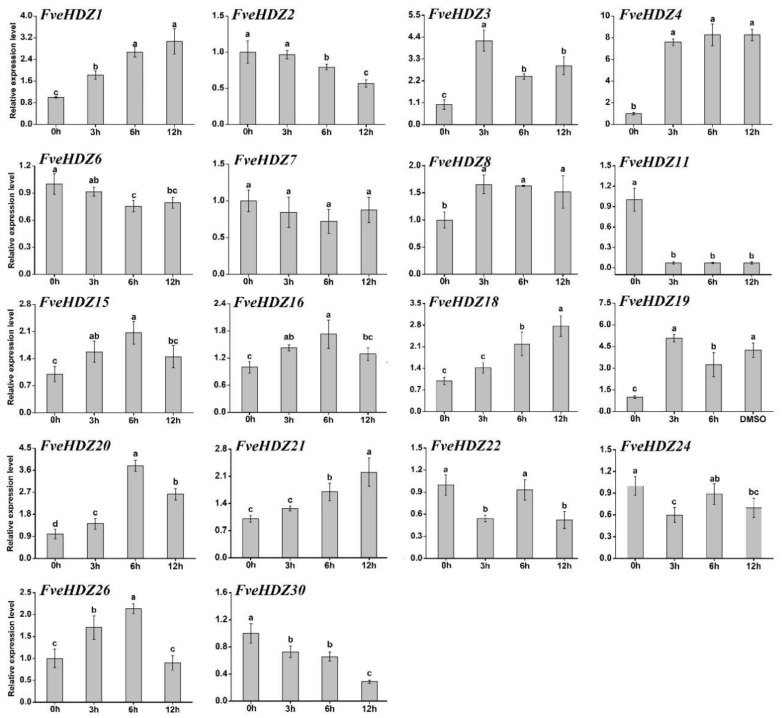
Relative expression levels of *FveHDZ* genes in leaves under ABA treatment. Expression at 0 h was normalized as “1” and used as control for each gene. Data were shown as mean ± standard deviation derived from three biological replicates, and different letters above the bars indicate significant differences based on Duncan’s multiple range tests (*p* < 0.05, n = 3).

**Figure 5 plants-11-03367-f005:**
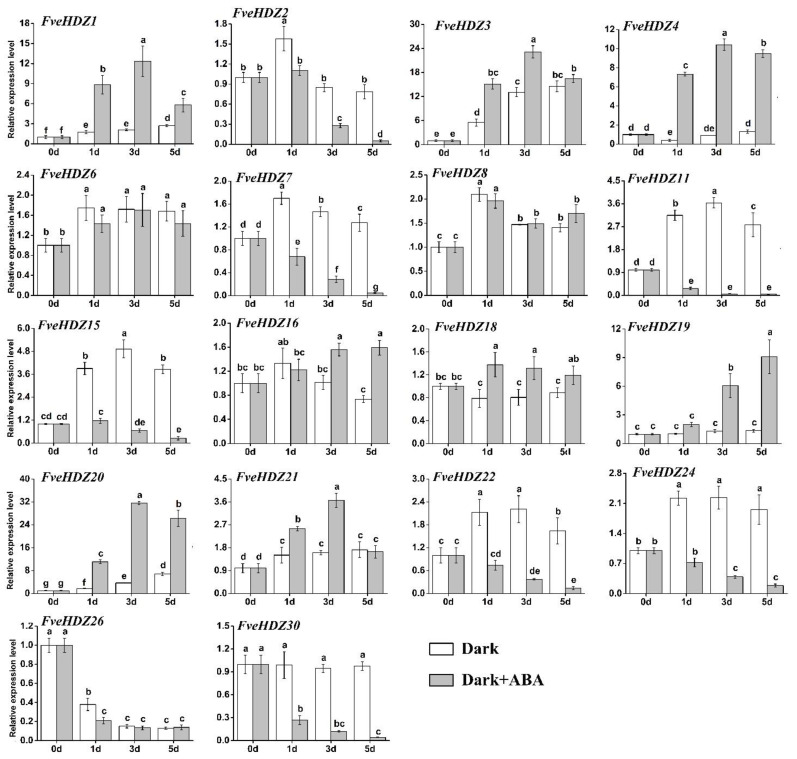
Relative expression levels of *FveHDZs* in the leaves in dark treated with (grey columns) or without (white columns) ABA. Expression at 0 d was normalized as “1” for each gene. Data were shown as mean ± standard deviation derived from three biological replicates, and different letters above the bars indicate significant differences based on Duncan’s multiple range tests (*p* < 0.05, n = 3).

**Figure 6 plants-11-03367-f006:**
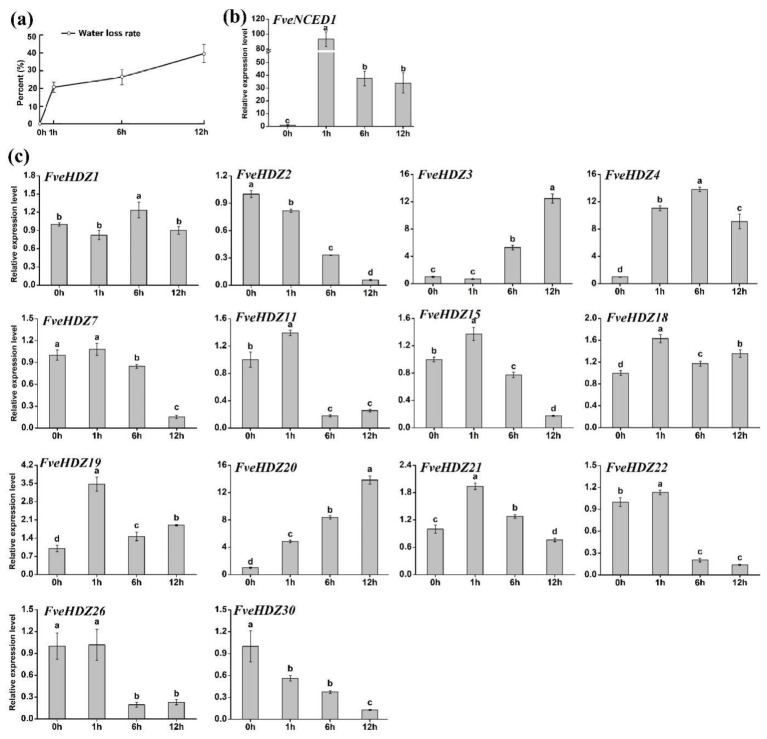
Water loss and relative gene expression levels during leaf dehydration. (**a**) The water loss rate of the dehydrating leaves. (**b**) Relative expression levels of *FveNCED1* (*FvH4_3g16730*), the key ABA synthesis gene in response to drought stress [45]. (**c**) Relative expression levels of *FveHDZs*. Expression at 0 h was normalized as “1” for each gene. Data were shown as mean ± standard deviation derived from three biological replicates, and different letters above the bars indicate significant differences based on Duncan’s multiple range tests (*p* < 0.05, n = 3).

**Figure 7 plants-11-03367-f007:**
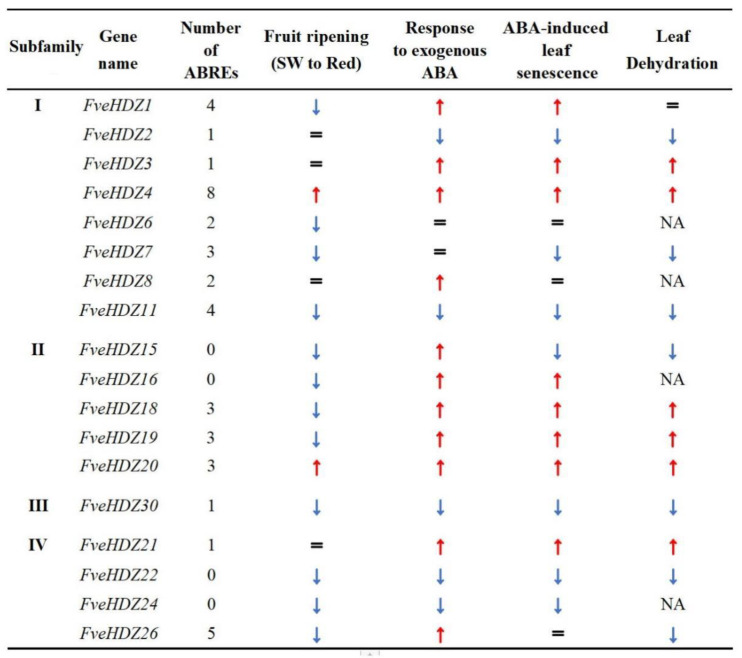
Summary of expression patterns of the selected 18 *FveHDZs* in the four ABA-mediated processes. Red “↑”, blue “↓”, and black “=“ indicate ≥1.5-fold up-regulated, ≥1.5-fold down-regulated, and almost unchanged expressions, respectively. Results of the fruit ripening process were based on the transcriptome data from Gu et al. [43] (Figure 3c). Results of the other three processes were based on the RT-qPCR analyses in this study (Figure 4, Figure 5 and Figure 6). NA, Not examined; ABRE, ABA-responsive element; SW, Small white stage.

**Figure 8 plants-11-03367-f008:**
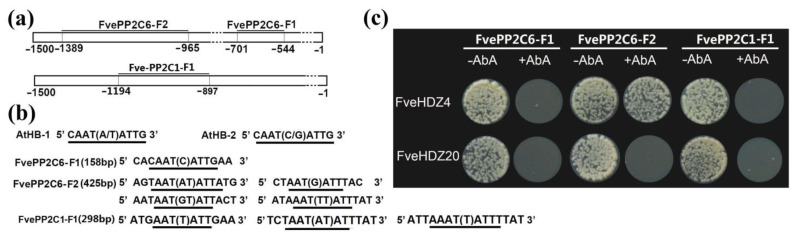
Possible interaction between HD-Zip proteins and *FvePP2C* promoters tested by yeast one-hybrid (Y1H) analysis. FveHDZ4 and FveHDZ20 proteins, as well as two fragments from the *FvePP2C6* promoter (FvePP2C6-F1 and FvePP2C6-F2, the upper in (**a**)) and one fragment from the *FvePP2C1* promoter (FvePP2C1-F1, the lower in (**a**)), were used in the Y1H assay (**c**). (**a**) Location of the selected fragments in the promoters of *FvePP2C6* (upper) and *FvePP2C1* (lower). The upstream 1500 bp region of the transcription start site (ATG) was shown as the promoter. (**b**) Binding sequences of *Arabidopsis* HD-Zip proteins ATHB-1 and ATHB-2 [4], and the putative FveHDZ-binding sequences found in FvePP2C6-F1, FvePP2C6-F2, and FvePP2C1-F1 (indicated with the underline). The length of each selected fragment as indicated in the parentheses behind the fragment name. (**c**) Yeast one-hybrid (Y1H) analysis between FveHDZ4/FveHDZ20 proteins and the *FvePP2C6* and *FvePP2C1* promoter fragments. No basal activity of these promoter fragments was detected on the medium containing Aureobasidin A (AbA, 200 ng/mL). –AbA and +AbA means without and with AbA in the medium, respectively.

**Table 1 plants-11-03367-t001:** Characteristics of the putative HD-Zip family members in *F. vesca*.

Gene Name	Subfamily	Intron No.	Protein Features					Duplication Type
			Protein Length (aa ^1^)	Molecular Weight (kDa)	Isoelectric Point (PI)	GRAVY ^2^	Subcellular localization	
FveHDZ1	I	2	286	32.78210	5.30	−1.076	Nucleus	WGD/segmental
FveHDZ2	I	2	329	37.20892	5.18	−0.881	Nucleus	WGD/segmental
FveHDZ3	I	1	228	26.55655	5.54	−1.125	Nucleus	WGD/segmental
FveHDZ4	I	1	243	27.91169	4.9	−1.095	Nucleus	WGD/segmental
FveHDZ5	I	2	237	27.45371	8.49	−1.033	Nucleus	dispersed
FveHDZ6	I	3	310	35.28906	4.84	−0.815	Nucleus	dispersed
FveHDZ7	I	2	327	37.14513	4.93	−0.835	Nucleus	WGD/segmental
FveHDZ8	I	2	326	36.83849	4.74	−0.802	Nucleus	WGD/segmental
FveHDZ9	I	2	320	35.90381	6.4	−0.932	Nucleus	WGD/segmental
FveHDZ10	I	2	304	34.48854	6.12	−0.846	Nucleus	WGD/segmental
FveHDZ11	I	0	172	19.76543	9.1	−0.701	Nucleus	dispersed
FveHDZ12	I	2	212	24.67798	9.2	−0.924	Nucleus	dispersed
FveHDZ13	I	2	215	24.78802	6.47	−0.886	Nucleus	dispersed
FveHDZ14	II	2	326	36.00138	6.24	−0.525	Nucleus	WGD/segmental
FveHDZ15	II	3	317	35.68198	8.48	−0.874	Nucleus	WGD/segmental
FveHDZ16	II	2	277	31.13987	8.43	−0.851	Nucleus	WGD/segmental
FveHDZ17	II	2	231	26.44407	8.84	−0.848	Nucleus	dispersed
FveHDZ18	II	3	342	38.01078	8.68	−0.695	Nucleus	dispersed
FveHDZ19	II	3	209	23.31446	8.94	−0.732	Nucleus	dispersed
FveHDZ20	II	2	253	28.40885	7.69	−0.872	Nucleus	WGD/segmental
FveHDZ21	IV	10	771	85.78192	5.73	−0.498	Nucleus	dispersed
FveHDZ22	IV	8	830	89.98074	5.94	−0.344	Nucleus	dispersed
FveHDZ23	IV	8	807	88.41103	5.41	−0.306	Nucleus	dispersed
FveHDZ24	IV	10	773	84.29501	5.55	−0.36	Nucleus	proximal
FveHDZ25	IV	10	777	85.43578	5.84	−0.299	Nucleus	WGD/segmental
FveHDZ26	IV	8	851	94.79136	5.68	−0.38	Nucleus	dispersed
FveHDZ27	IV	9	709	78.56529	6.43	−0.322	Nucleus	dispersed
FveHDZ28	III	17	844	92.98051	5.94	−0.134	Nucleus	dispersed
FveHDZ29	III	17	847	92.77479	6.07	−0.138	Nucleus	dispersed
FveHDZ30	III	17	845	92.88056	6.06	−0.148	Nucleus	dispersed
FveHDZ31	III	17	838	91.87144	6.03	−0.086	Nucleus	dispersed

^1^ amino acid residues; ^2^ Grand average of hydropathicity.

## Data Availability

Not applicable.

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
