# Peer review of "Genome-Wide Characterization and Expression Profiling of HD-Zip Genes in ABA-Mediated Processes in Fragaria vesca"

_plants, 2022, doi:10.3390/plants11233367_

Round 1

Reviewer 1 Report

In my opinion, the work is interesting and multifaceted. There is no

doubt that transcription factors are the basis of the regulation, so the

significance of the work is undeniable. The work is a very successful

fusion of the bioinformatic approach (multilateral analysis of databases

with the most modern algorithms) and the experimental approach (RT-PCR

under conditions of exogenous ABA, darkness and drought). The work is

written in an understandable language, well illustrated, the results are

discussed in detail and competently. Undoubtedly, the work will be of

interest to a wide range of readers, and deserves publication in Plants. 

In Fragaria vesca, which is a model object for studying the genus Fragaria, was revealed 31 homeodomain-leucine zipper transcription factors genes. These TFs are involved in such ABA-mediated processes as fruit ripening, dehydration, leaf senescence and stress response.Knowledge of the interaction of these factors is necessary to understand the course of these processes.An extensive study of the activity of these genes under conditions of exogenous ABA, dehydration and leaf senescence was carried out using RT-PCR.The obtained results allow us to detail the main ABA-mediated processes in strawberry and other plants. This topic is undoubtedly relevant in this area.

HD-Zip TFs are relatively poorly studied in plants, with data available only for Arabidopsis, rice an populus.This work undoubtedly expands our understanding of them. The work was carried out at a high methodological level.The latest algorithms were used for bioinformatic analysis, and modern RT-PCR was used for experimental analysis. The conclusions are fully consistent with the results obtained and shed light on the main question posed. The references  are appropriate. The figures are well done and clearly help to understand the results.

Author Response

Response : Thank you for your kind and positive evaluation of our manuscript. We have revised the manuscript according to the comments of the other reviewers.

Reviewer 2 Report

The manuscript reports genome-wide identification and expression analysis of HD-Zip genes from Fragaria vesca. FveHDZ4 and 20 were focused on based on the results of expression analysis. Yeast one-hybrid (Y1H) assay suggests that FveHDZ4 regulates FvePP2C6 expression. However, additional experiments are needed to prove the regulation of FvePP2C6 by FveHDZ4, as described below.

1) The mutation analysis for putative binding sites of FveHDZ4 on the FvePP2C6 promoter should be performed in the Y1H assay.

 2) Expression analysis of FvePP2C6 would better be performed under the conditions shown in Figures 4, 5, and 6; if FvePP2C6 is under the control of FveHDZ4, FvePP2C6 expression might be induced after FveHDZ4 induction.

Reviewer 3 Report

This paper was trying to generalize the common gene features of the HD-Zip Genes family in Fragaria vesca. Although that information on features is basically and procedurally collected from the public database, the author simultaneously investigates the expression response of these gene members under a couple of varied ABA-mediated treatments. The study seems quite conventional but shows a general characterization of such gene family in the Fragaria vesca which may probably be a valuable reference for relative research. The whole paper was well organized, and the writing seems fluent. I have several following concerns:

1. From the view of molecular evolution, gene origination is actually quite a traceable process, in Table 1, the Duplication type of all the genes seems not well supported by the evidence offered by the author. The author mentioned “syntenic regions in the F. vesca genome were analyzed by the MCScanX software”,  however, the author should further confirm whether MCScanX has adequate power to confirm the WGD events since it’s impossible that every gene is based on duplication (should have one/several orthologous genes from ancestral sequence/gene before WGD).

2. Figure 1:didn’t indicate the Ks or Ka and the type of phylogenetic tree.

3. It’s easy to get the point that many HD-Zip family members are involved in ABA responses in the view of EVO-DEV, however, the result induced an interesting  hesitation that why some of the gene family members are up-regulated while some others are down-regulated. The author should at least discuss this. This will be the initial clue to figuring out a comprehensive scene of the genetic mechanism of ABA response in terms of the HD-Zip genes family.

4. Tech issue: for RT-qPCR, the author didn’t clear how they avoid the non-specific issue for the design of the RT-qPCR primers and whether they’ve pre-tested the primes amplification efficiency.

5. For the study of the gene family, people seem seldomly trace the gene age of the big family member, investigating the expanding of the gene members in the population/species level, or even to compare between different gene families, the reason may be that it’s hard even to exactly classify some of the doubtful homologous. However, for this case: the HD-Zip genes family, would it be the best gene family model to figure out some basic questions: how does the gene family expand to the current scenario and why?.

Round 2

Reviewer 2 Report

The reviewer understood the response regarding the revision of the discussion for the Y1H assay. On the other hand, the reviewer thinks that the correlation between HDZ4 and PP2C6 expression is not adequately addressed. In Fig. S5, PP2C6 expression is only shown in the ripening stage. Does PP2C6 expression correlate with HDZ4 in the other stages shown in Fig. 3? Even if it is impossible to perform qRT-PCR experiments, the correlation of expression should be analyzed with previous transcriptome data.
